

# Potential tsunami hazard of the southern Vanuatu Subduction Zone: tectonics, case study of the Matthew Island tsunami of 10 February 2021 and implication in regional hazard assessment.

Jean Roger[1,*], Bernard Pelletier[2], Aditya Gusman[1], William Power[1], Xiaoming Wang[1], David Burbidge[1], Maxime Duphil[3,4]

[1]GNS Sciences, 1 Fairway Drive, Lower Hutt 5010, New Zealand

[2]GEOAZUR, Institut de Recherche pour le Développement, 101, Promenade Roger Laroque, BP A5 98848 Nouméa CEDEX, New Caledonia

[3]ENTROPIE, Institut de Recherche pour le Développement, 101, Promenade Roger Laroque, BP A5 98848 Nouméa CEDEX, New Caledonia

[4]École Doctorale n°129 Sciences de l'Environnement, UPMC Sorbonne Université, 4, Place Jussieu, 75005, Paris CEDEX, France.

*Correspondence to:* J. Roger (jean.roger@gns.cri.nz)

## Abstract

The Vanuatu subduction zone (VSZ) is known to be seismically very active, releasing a significant energy resulting of the quick convergence rate between the Australian and Pacific tectonic plates. That is not the case on its southernmost part south of latitude 22.5°S and east of longitude 170°E which is neither known as being highly tectonically active nor having produced large tsunamis over the past 150 years, and by the way, has not been much studied. On the 11[th] of February 2021 (10 February UTC), a magnitude $M_w$ 7.7 earthquake triggered a tsunami warning in New Caledonia and Vanuatu twenty minutes after midnight (local time). With an epicentre located close to the volcanic islands of Matthew and Hunter, this shallow reverse-faulting rupture (< 30 km depth) was able to disturb the seabed and produce a tsunami. In fact, it was confirmed 45 min later by the coastal gauges of the Loyalty and the south Vanuatu islands which recorded the first tsunami waves. Showing an overall recorded amplitude of less than 1 m with a maximum of ~1.5 m in Lenakel, (Tanna, Vanuatu), it has been recorded on most coastal gauges and DART stations of the southwest Pacific Region as far as Tasmania in the South and Tuvalu in the North respectively at distances of ~3000 and ~1800 km from the epicentre. In this study, the tsunamigenic potential of the southernmost part of the VSZ and the implications in terms of regional





hazard assessment are discussed through (1) the presentation of the complex tectonic settings of this
"transition zone" between the Solomon-Vanuatu and the Tonga-Kermadec Trenches; (2) the case study
of the 10 February 2021 tsunami at a southwest Pacific regional scale using three different tsunami
generation scenarios computed with COMCOT modelling code on a set of 48 nested bathymetric grids;
and (3) the simulation of a plausible $M_w$ 8.2 scenario encompassing the active part of this "transition
zone". In fact, the validation of the $M_w$ 7.7 parameters for tsunami modelling provides keys to further
assess the hazard from potential tsunami triggered by higher magnitude earthquakes in this region.
Finally, it helps to highlight the significant role played by the numerous submarine features in the
region, the Norfolk Ridge being the most important acting like a waveguide toward the north and the
south.

**Keywords:** tsunami numerical modelling, Vanuatu-New Hebrides subduction zone, earthquake,
Matthew Island

# 1. Introduction
### 1.1 Generalities
On 10 February 2021 at 13:19:55 UTC (11 February at 00:19:55 LT) a $M_w$ 7.7 earthquake occurred at
the southernmost part of the Vanuatu Subduction Zone (former New Hebrides Subduction Zone; called
VSZ in the rest of this article), 420 km from Maré, Loyalty Islands, New Caledonia and ~80 km from
the two small uninhabited volcanic islands of Matthew and Hunter, respectively located at 171.35°E
and 22.34°S and 172.07°E and 22.4°S (**Figure 1**). While this earthquake was only felt by a few people
in New Caledonia and Vanuatu because it occurred far away from the inhabited islands and during the
night, it was quickly followed by a regional tsunami warning provided by the Pacific Tsunami Warning
Centre (PTWC) and the New Zealand National Geohazards Monitoring Centre (NGMC). From 45
minutes after the shaking, a tsunami was recorded by the coastal gauges located along the coast of New
Caledonia and Vanuatu, and later along the northern coast of New Zealand, Norfolk Island, the eastern
coast of Australia and most of the coastal gauges located in the southwest Pacific Ocean.



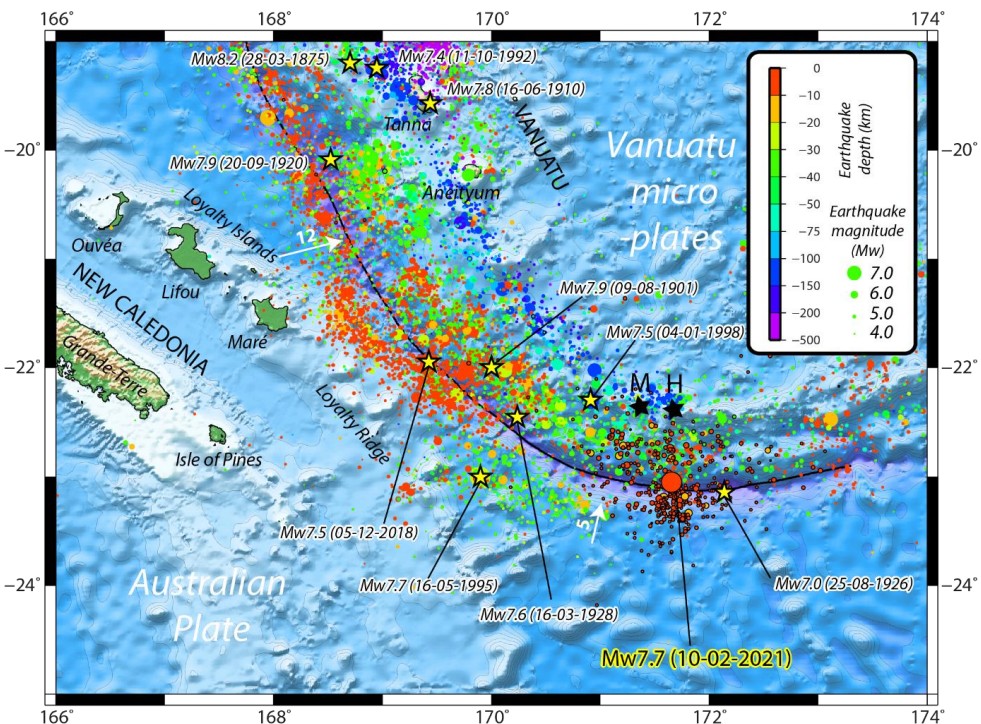


**Figure 1: Local seismotectonic context: location of the 10 February 2021 M$_w$ 7.7 earthquake at the interface between the Australian Plate and the Matthew-Hunter micro-plate (part of the Vanuatu micro-plates complex, southernmost Vanuatu arc). Earthquakes (M$_w$ > 3.0) from USGS from 1 January 1970 to 31 March 2021 are shown by coloured circles, those with a black outline being recorded from the 10[th] of February to 31[st] of March. Convergence rates (in cm/yr) are represented by the white arrows. Yellow stars locate strong historical earthquakes. Black line symbolizes the subduction trench. The two black stars locate Matthew (M) and Hunter (H) islands. Topographic data extracted from GEBCO2021 dataset (VLIZ/IOC, 2021).**

**1.2 Objectives of this study**

From a hazard assessment perspective, this study aims to understand what did happen in this relatively inactive part of the VSZ (1) discussing the complex seismotectonic context; (2) using numerical simulations of the 10 February 2021 tsunami generation and propagation in the southwest Pacific Ocean: three scenarios are tested for tsunami generation going from a simple uniform slip model prepared with seismic data and empirical relationships between fault parameters, the USGS finite fault model provided for this earthquake, and a subsequent waveforms inversion of the signal recorded at



New Zealand DART and coastal gauges; (3) the simulation results help to propose a plausible $M_w$ 8.2
earthquake rupture scenario and simulate its propagation in the southwest Pacific region. Notice that all
the dates and times are in UTC in the rest of the article.

**2. Seismotectonic context**
The VSZ (10-23°S, 165-173°E), including from south to north the French Matthew and Hunter volcanic
islets, Vanuatu and Eastern Solomon Islands, is among the world's fastest moving plate boundaries with
a convergence rate of up to 16-17 cm/y in the northern part between the Australian Plate on the west
and several Vanuatu micro-plates on the border of the Pacific Plate to the east (Louat et Pelletier, 1989;
Pelletier et al., 1998; Calmant et al., 2003). It has a history of producing numerous moderate to strong
earthquakes (Louat and Baldassari, 1989; Cleveland et al., 2014; Ioualalen et al., 2017). The largest
events recorded during the instrumental period (since 1900) have moment magnitudes of between $M_w$
7.8 to 8.0 and are located in both the northern ($M_w$ 7.8 on 7 October 2009 and $M_w$ 8.0 on 6 February
2013 events) and the southern parts ($M_w$ 7.9 on 9 August 1901, $M_w$ 7.9 on 20 September 1920 and $M_w$
7.9 on 2 December 1950 events) of the subduction zone. However, the maximum magnitude of
earthquakes on the zone may be higher, the moment magnitude of the 28 March 1875 earthquake in the
southern part having been estimated to $M_w$ 8.1-8.2 (Ioualalen et al., 2017). Note that there are some
questions raised about the 9 August 1901 earthquake location (-22°, 170°) and magnitude: it goes from
$M_w$ 7.9 to 8.4 according to Gutenberg (1956), Richter (1958) and Engdahl and Villasenor (2002) but it
has not been reported in the highly detailed earthquakes catalogue of New Caledonia from Louat and
Baldassari (1989). By contrast, no large thrust events have been recorded in the central part (between
14°S and 17°S), the maximum recorded magnitude being $M_w$ 7.6 on 11 August 1965, and especially in
the southernmost part of the subduction zone (south 22.5°S and east of 170°E), with a maximum
magnitude of $M_w$ 7.0 on 25 August 1926 (see **Figure 1** for earthquakes location).
Calmant et al. (2003) estimated the convergence rate on the subduction zone to the south of the
Matthew-Hunter Islands to be ~45 mm/yr. This value has been confirmed by Power et al (2012) who
obtained 46-48mm/yr in their best fitting elastic block model requiring minimal interseismic coupling





(less than about 0.2). However, the large uncertainties on GPS data meant that it was not possible to
constrain the degree of coupling in this area with any accuracy (Power et al, 2012).  If the coupling was
indeed this low, it would suggest that the seismicity expected in this area would be much lower than
expected for a zone with this rate of convergence.
The area of the southern part of the VSZ between the latitudes 21.5° and 22.5°S and the longitudes 169°
and 170°E is very active seismically and has produced several seismic crises with earthquakes of
magnitude $M_w$ 7.0+ during recent decades (1980, 1995, 2003-2004, 2017, 2018). These events are felt
by the population in New Caledonia and Vanuatu as discussed by Roger et al. (2021). From a geological
point of view, this region is characterized by the progressive subduction/collision of the NW-SE
trending Loyalty Ridge located on the Australian Plate under the southern Vanuatu micro-plates. This
leads to strain accumulation that is regularly partially released through moderate to strong earthquakes
during remarkable sequences (1980, 2003-2004, 2017, 2018) which include both interplate thrust
faulting earthquakes and outer rise normal faulting earthquakes west and southwest of the trench, in
which events of one mechanism appear to trigger events of the other (Roger et al., 2021).
The subduction/collision of the Loyalty Ridge is considered to have a large influence on the local
tectonics, on both the overthrusting and the subducting plates (Louat et Pelletier, 1989; Pelletier et al.,
1998; Calmant et al., 2003). Northwest of the Loyalty Ridge and trench junction (southern part of the
VSZ) the GPS-derived convergence is 12 cm/y and is trending ENE-WSW while southeast of the
junction (22°S) the convergence is reduced (5 cm/y) and is almost N-S in front of Matthew-Hunter
islands, implying a large (9 cm/y) left lateral motion and/or NW-SE extension in the upper plate along
or at the rear of the Matthew-Hunter islands (**Figure 2**) as also shown by numerous strike slip and NE-
SW trending normal faulting events. The region is thus potentially able to trigger tsunamis with a main
propagation axis striking from WSW-ENE (potential main energy path towards New Caledonia and
south Vanuatu) to S-N (potential main energy path toward New Zealand and Vanuatu). Deformation of
the subducting plate is well illustrated by the seismicity and the focal mechanism solutions of normal
faulting earthquakes on the outer rise of the trench, which follow the bend of the trench (**Figure 2**).
From north to south these outer rise events are distributed along three lineations trending WNW-ESE,



NW-SE and almost W-E, and located further and further from the trench, suggesting a twist of the plate.
The largest normal faulting earthquake ($M_w$ 7.7 on 16 May 1995) is located on this southern lineament
which in detail includes three segments and strikes almost E-W toward the Isle of Pines in southern
New Caledonia. Possibly the seismicity in the southern part of the Grande Terre and the south lagoon
of New Caledonia (showing $M_w$ 5.6 normal faulting and $M_w$ 5.1 strike slip faulting earthquakes
respectively on December 1990 and February 1991) may result from stress induced by the ongoing
subduction of the Loyalty Ridge at the southern end of the VSZ.
From a tsunami generation point of view, the VSZ is not as known as other subduction zones to trigger
catastrophic tsunamis able to strongly impact coastal communities. According to recent catalogues of
tsunamis in New Caledonia (Sahal et al., 2010; Roger et al., 2019a), only 16 of the 37 (17 of the 38 if
including the 10 February 2021 tsunami) have been generated at the VSZ since 1875 and amongst them,
5 show a maximum recorded/reported amplitude > 50 cm. The ratio 5/17 is to consider with caution:
most of the small tsunamis have been recorded by coastal gauges (but not reported by witnesses) during
the last decade and thus, the real number of tsunamis having reached New Caledonia, at least from the
VSZ, is probably considerably bigger than 17. The latest earthquake-generated tsunami triggered by the
VSZ occurred on 5 December 2018, following an $M_w$ 7.5 normal faulting earthquake (Roger et al.,
2019a,b; Roger et al., 2021): its amplitude reached more than 2 m in some locations in the south of New
Caledonia and Vanuatu. (Note: at the time of the article submission, there is at least 2 new tsunamigenic
earthquakes of magnitude $M_w$ 6.9 and 7 having occurred on the VSZ on 30 and 31 March 2022)

## 3. Case study: the 10 February 2021 earthquake and tsunami

### 3.1 The earthquake

The 10 February 2021 $M_w$ 7.7 earthquake, located around 23°S, 171.6°E, 170 km east of the 1995 $M_w$
7.7 earthquake, hitherto known to be the strongest recorded earthquake in southernmost VSZ, is
interesting in the sense that it occurred nearly at the southeasternmost part of the trench, with a
magnitude much stronger than the usual low seismicity previously recorded in this region (**Figure 1**



and **Figure 2**). Indeed, the prior and closest main event in this area was the 25 August 1926 $M_w$ 7.0
earthquake, located at 23.14°S, 172.14°E, about 60 km further east. The epicentre being closer to
Matthew Island than Hunter Island, the name "Matthew Island earthquake" was retained in the
aftermath of the event.
The $M_w$ 7.7 main shock was preceded by 13 foreshocks with notably six events ($M_w$ 5.1 to 5.8) in one
hour on February 2-3 and three events ($M_w$ 5.8 to 6.1) within the hour before the mainshock. All the
main foreshocks have similar focal mechanism solutions to the main shock, i.e. thrust faulting, as shown
with the moment tensor solutions (GCMT project: Dziewonski et al., 1981; Ekström et al., 2012) on
**Figure 3**. Almost 100 aftershocks of magnitude $M_w$ 5+ have occurred after the main shock.

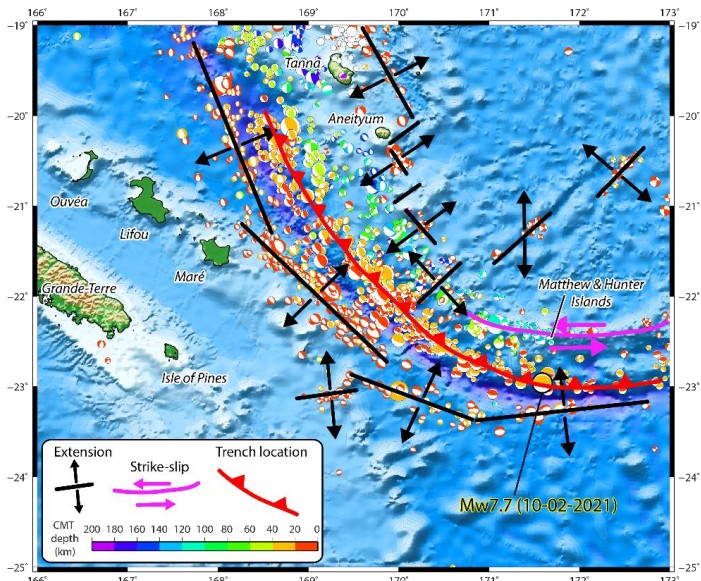


**Figure 2: Focal mechanisms from the GCMT project in the southern part of the Vanuatu Subduction Zone and**
**geodynamical interpretation.**


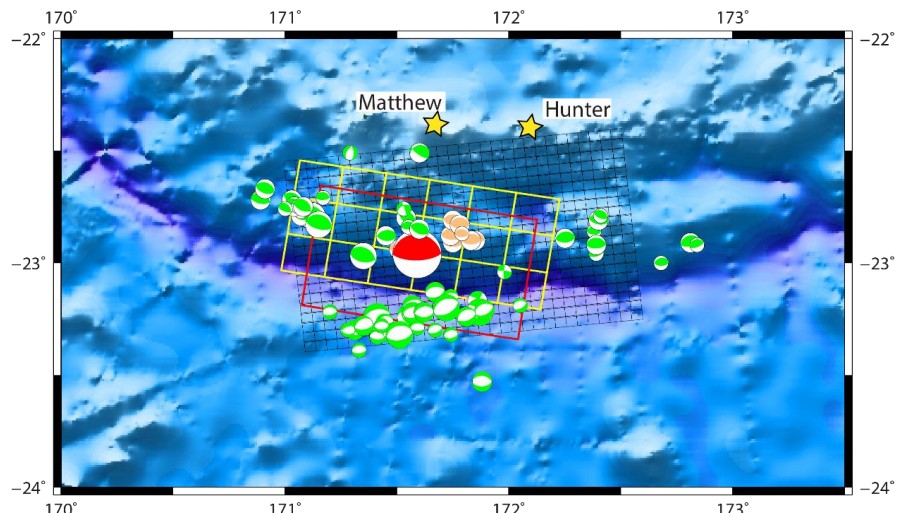


**Figure 3 : Map of the centroid moment tensors (GCMT project; last accessed on 10 May 2022) calculated for the main earthquakes (Mw ≥ 5) occurring during the February 2021 seismic crisis (from 1 to 28 February) south of Matthew and Hunter islands (yellow stars). Red colour stands for the main shock, orange for the foreshocks and green for the aftershocks. The extent and the number of subfaults of the 3 scenarios used in this study is represented by the black, yellow and red rectangles standing respectively for the USGS finite fault model, the non-uniform model obtained from tsunami waveforms inversion and the uniform slip model.**

According to the focal mechanism solutions provided by USGS (https://earthquake.usgs.gov/earthquakes/eventpage/us6000dg77/moment-tensor), GCMT (https://www.globalcmt.org), GEOSCOPE-IPGP-Scardec (http://geoscope.ipgp.fr), French Polynesian Tsunami Warning Center (cppt@labogeo.pf) and GFZ Geofon (http://geofon.gfz-potsdam.de/eqinfo), this earthquake exhibits a nearly pure compression mechanism (reverse faulting event with a small strike-slip component) and likely occurred at the subduction interface on a shallow (depth ranges from 12 to 29 km depending of the agencies: 25.5 km (USGS) and 21.8 km (GCMT)) fault striking parallel to the trench (strike ranges from 246° to 281° (USGS and GCMT strike of 246° and 279° respectively) as shown on **Figure 3**, and dipping to the north (dip ranges from 11 to 27°: 17° (USGS) and 23° (GCMT)).

**3.2 Fault slip models**





Within the framework of the present study, three different rupture scenarios have been considered to
simulate initial seafloor displacement: 1) a uniform slip model; 2) a non-uniform slip model obtained
with inversion of tsunami waveforms; 3) a non-uniform slip model obtained with inversion of seismic
and GPS data. An additional uniform slip scenario is proposed for further discussion on tsunami hazard
consideration from this region of the VSZ. (Note that the authors are aware of the recent publication of
Ye et al. in December 2021 proposing another finite-fault slip model from inversion of teleseismic body
waves)
*3.2.1 Uniform slip model (scenario #1)*
GCMT, Geoscope and the USGS calculated the seismic moment associated to the earthquake of
respectively $M_0 = 4.01 \times 10^{20}$ N.m, $M_0 = 4.25 \times 10^{20}$ N.m, and $M_0 = 4.364 \times 10^{20}$ N.m. It corresponds to
a magnitude $M_w = 7.67$ to 7.69 according to $M_w = \frac{2}{3}\log_{10}(M_0) - 10.73$ (Hanks and Kanamori, 1979)
where $M_0$ is in dyne.cm. Geoscience Australia estimated the moment magnitude to be slightly lower
($M_w = 7.61$).
In this study, a uniform slip scenario has been built based on the GCMT solution
(https://www.globalcmt.org), which is generally better than other solutions in terms of epicentre
location and fault azimuth correlated with existing features for earthquakes located at the VSZ and
nearby. For this purpose, it is assumed that the rigidity coefficient is $\mu = 3 * 10^{11}\ dyn.cm^{-2}$
corresponding to a depth of 22 km (Bilek and Lay, 1999). According to the empirical relationships of
Blaser et al. (2010) and Strasser et al. (2010) the length L and width W of the fault plane have been
respectively calculated to 100 km and 60 km. To match with the GCMT seismic moment this
corresponds to an average coseismic displacement on the fault plane S = ~2.2 m. The parameters
determined for the uniform slip modelling are summarized in **Table 1**.




**Table 1: Parameters used for the initial deformation calculation associated to uniform slip ruptures corresponding to M$_w$ 7.7 and M$_w$ 8.2 earthquakes.**

|  | Lon (°) | Lat (°) | Depth (km) | Length (km) | Width (km) | Strike (°) | Dip (°) | Rake (°) | Slip (m) |
|---|---|---|---|---|---|---|---|---|---|
| Simple fault plane M$_w$7.7 | 171.59 | -22.96 | 21.8 | 100 | 60 | 279 | 23 | 101 | 2.2 |
| Simple fault plane M$_w$8.2 | 171 | -22.8 | 25 | 220 | 80 | 287 | 20 | 90 | 5.0 |

### 3.2.2 Non-uniform slip model (scenario #2)

The observed tsunami waveforms recorded at 4 DART and 24 coastal stations were used in a tsunami waveforms inversion to estimate the fault slip distribution of the 2021 Loyalty Island earthquake (Gusman et al., *in revision*). The geometry for the fault model was based on the GCMT solution. The estimated slip distribution has a major slip region with maximum slip amount of 4.1 m located near the trench, this estimated large slip near the trench being consistent with the fault slip model estimated by the USGS (see section 3.2.3). The estimated maximum uplift near the trench is 2.1 m while the subsidence is 0.24 m. The previous study by Gusman et al. (*in revision*) used an assumed rigidity of 4 $\times$ 10$^{10}$ N.m$^{-2}$ to get a seismic moment of 3.39 $\times$ 10$^{20}$ N.m (M$_w$ 7.65) from the estimated slip distribution. However, if we assume the rigidity to be of 3 $\times$ 10$^{10}$ N.m$^{-2}$, the calculated seismic moment of the fault slip model would be 2.54 $\times$ 10$^{20}$ N.m (M$_w$ 7.57), which is smaller than those calculated by GCMT and USGS.

### 3.2.3 USGS finite fault model (scenario #3)

In the aftermath of the main shock, the USGS has released a kinematic finite fault model of the rupture (https://earthquake.usgs.gov/earthquakes/eventpage/us6000dg77/finite-fault) calculated from inversion of seismic and GPS data with an approach based on Ji et al. (2002)'s methodology.

The resulting model is composed of 620 5km-by-5km sub-segments. Each segment has its own depth, slip, rake and rupture time values. The file used in this study is available here: https://earthquake.usgs.gov/archive/product/finite-fault/us6000dg77_1/us/1613004810949/basic_inversion.param [Last accessed in February 2021].

### 3.2.4 Plausible M$_w$ 8.2 uniform slip model (scenario #4)

This scenario is based on the fact that the southernmost part of the VSZ (east of 170°E) has not experienced any strong earthquake for at least 100 years, exhibiting a shortening of at least 5 m corresponding to a convergence rate of 5 cm/yr, enabling it to produce easily a magnitude $M_w$ 8.0-8.2, according to the length of active plate boundary available here. The same empirical relationships as in the scenario #1 have been used to set up the corresponding parameters of a $M_w$ 8.2 rupture: pure thrust mechanism (rake = 90°) with 5 m displacement on the fault plane, length, width and depth of the fault plane of respectively 220 km, 80 km and 25 km, an azimuth of 287°, a dip of 20°. The epicentre of the rupture is chosen at 171°E, 22.8°S. The parameters are summarized in **Table 1**. Note that this scenario does not consider a possible rupture of the VSZ toward the north, between the Loyalty Islands and Vanuatu, which would potentially lead to stronger magnitude earthquake. Also, due to the bending of the VSZ, this scenario represents only one of many possibilities of rupture energy directivity with a mean strike value on a pure thrust rupture aiming to discuss what could happen with a stronger magnitude than the February 2021 earthquake: depending on the strike, the rake and the epicentre location, the main energy paths would probably completely change the directivity pattern of the tsunami. A more accurate study would consider incorporating the shape of the subduction interface as proposed with the SLAB 2.0 model (Hayes, 2018) using for example a triangular mesh of the source, with variations of the strike, rake, and eventually, different slip distributions and a rupture time pattern.

### 3.3 The tsunami

The tsunami triggered by the 10 February 2021 earthquake can be classified as a region-wide event as it was recorded at least on 31 coastal gauges and 4 DART stations in the southwest Pacific, firstly on those of New Caledonia and Vanuatu, but also in Fiji, New Zealand (~1200 km), Australia (~1800 km) and as far as Tasmania (~3000 km) in the south and Western Samoa (~2000 km) in the east. For the purpose of this study the records of those gauges have been downloaded from LINZ website for what concerns the New Zealand coastal gauges network (https://www.linz.govt.nz/sea/tides/sea-level-data/sea-level-data-downloads [Last accessed in February 2021]) and on the IOC website (VLIZ/IOC, 2021) for other regional gauges. The New Zealand DART data are now publicly available on





https://www.geonet.org.nz/tsunami/dart [Last accessed on 31 May 2022]. They are shown on **Figure 4**
in a chronologic order and they represent the sea-level fluctuation with a sample rate of 1 min (coastal
gauges) and 15" (DART stations). **Figure 4** also shows the arrival of the tsunami at different times of
the tide from one station to another one. **Figure 5** shows the location of the coastal gauges and New
Zealand DART stations having recorded the tsunami. In good agreement with the tsunami travel times
(TTT) computed with Mirone software (Luis, 2007) on a 30" GEBCO grid also shown on **Figure 5**, it
was first recorded on MARE (Tadine, Maré, Loyalty Islands, New Caledonia)'s coastal gauge and LIFO
(Wé, Lifou, Loyalty Islands, New Caledonia) at 14:06 UTC, 46 minutes after the earthquake, shortly
followed by LENA (Lenakel, Tanna, Vanuatu) at 14:16 UTC. Meanwhile, the tsunami propagated
towards the south/south-west and reached KJNI (Norfolk Island, Australia)'s coastal gauge at 14:44
UTC, NCPT (Cape North, New Zealand)'s tsunami gauge at 15:26 UTC and finally SPJY (Southport)
and BAPJ (Battery Point) in Tasmania, Australia's southernmost coastal gauges, at 19:31 and 20:35
UTC respectively, 6 hours and 12 minutes and 7 hours and 16 minutes after the earthquake. Also, it was
recorded to the east on VITI and LEVU (Suva and Lautoka, Viti Levu, Fiji)'s coastal gauges at ~14:49
and ~15:17 UTC respectively, UPOL (Apia, Upolu Island, Western Samoa) at ~16:51 UTC, NKFA
(Nuku'alofa, Tonga) at ~16:29 UTC and in the north at FONG (Fongafale, Tuvalu) at ~16:25 UTC. Its
general maximum amplitude of less than 1 m classifies it in the small tsunamis' category but
nevertheless, it exhibited a stronger maximum amplitude of ~1.3 m recorded on LENA (Lenakel, Tanna,
Vanuatu). The tsunami arrival times and amplitudes at each coastal gauge and DART station are
summarized in **Table 2**. They have been obtained through de-tiding and filtering of the data using the
methodology presented in Roger (*Subm.*). Higher amplitudes can be expected in nearby exposed areas
showing particular geometries like V-shape bays, harbours and river mouths or specific submarine
features like submarine canyons and seamounts able to trigger amplification and/or resonance effects
of the incoming tsunami like it was the case on 5 December 2018 (Roger et al., 2021). At the regional
scale, its amplitude is higher close to the source region (New Caledonia, Vanuatu) and in the
southwestern quadrant (New Zealand, Australia). It is worth noting that the delay between the first wave
arrival and maximum amplitude reached by the tsunami has a median value of 1 hour and 24 minutes,
with a minimum delay of 8 minutes (the maximum amplitude recorded on DART NZG corresponds to



the first wave recorded on this DART) and a maximum delay of 7 hours and 24 minutes (NAPT, Napier, New Zealand).

Four of the six newly deployed New Zealand DART sensors were able to record the 10 February 2021 tsunami, arriving on DART NZE first, followed by NZG, NZC and NZI. Their records are shown on **Figure 4** and the stations are located on **Figure 5**, the related tsunami arrival times and amplitudes being summarized in **Table 2**. In each case, the record shows high frequency waves arriving a few minutes after the earthquake which are directly linked to the bottom shaking from internal seismic waves. This is particularly highlighted on the wavelet's spectrograms computed for each record (**Figure 6**). This is followed by lower frequency waves probably linked to the surface seismic waves (for more details about seismic wave records on DART, see Kubota et al., 2020). Then, between 2 and 3 hours after the main shock, the tsunami wave train is recorded showing a leading wave period of ~15 to 20 min depending on the azimuthal location of the DART station relatively to the strike of the fault: the closer the DART station is to the azimuth direction of the fault, the larger the period is.

It is important to notice that at the time of the earthquake the southwest Pacific Ocean was subject to one tropical storm (named 20P) south of Tonga and Fiji and a second storm located south of New Zealand and affecting some coastal gauge records with a wide range of frequencies. As underlined by Thomson et al. (2007) during the 2004 Sumatra tsunami or more recently by Roger (*Subm.*) for the March 2021 Kermadec tsunami, the frequency content of the storm generated waves possibly overlaps the tsunami signal, being able to show periods of several minutes. This is particularly the case for the Puysegur gauge (PUYT) as shown on **Figure 7**.


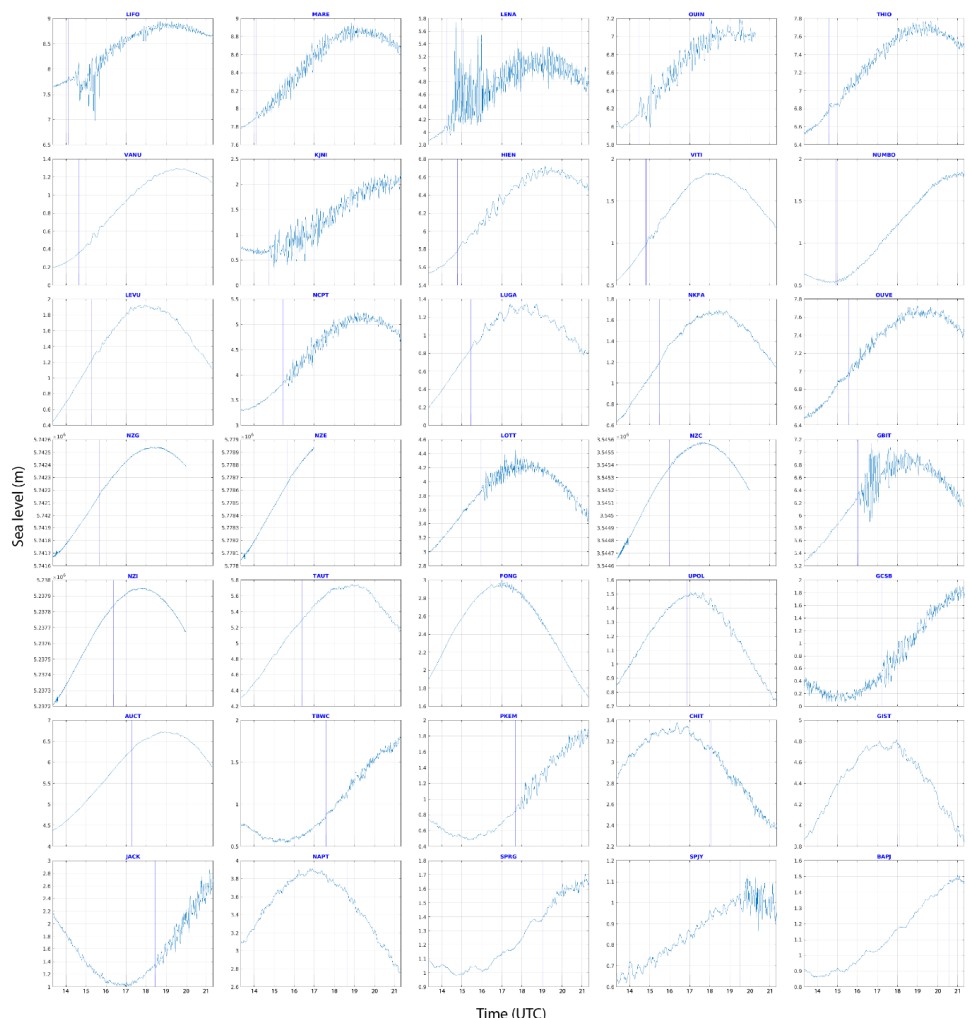

**Figure 4: 35 coastal gauge and New Zealand DART station records of the 10 February 2021 tsunami in the southwest Pacific Ocean. Each record begins at the earthquake time and goes for 9 hours. The vertical blue line symbolizes the tsunami arrival time (reported in table 2). For the 4 DART records, only the high-resolution signal (15" sampling rate) transmitted in real-time by the BPR to the monitoring centre is plotted.**
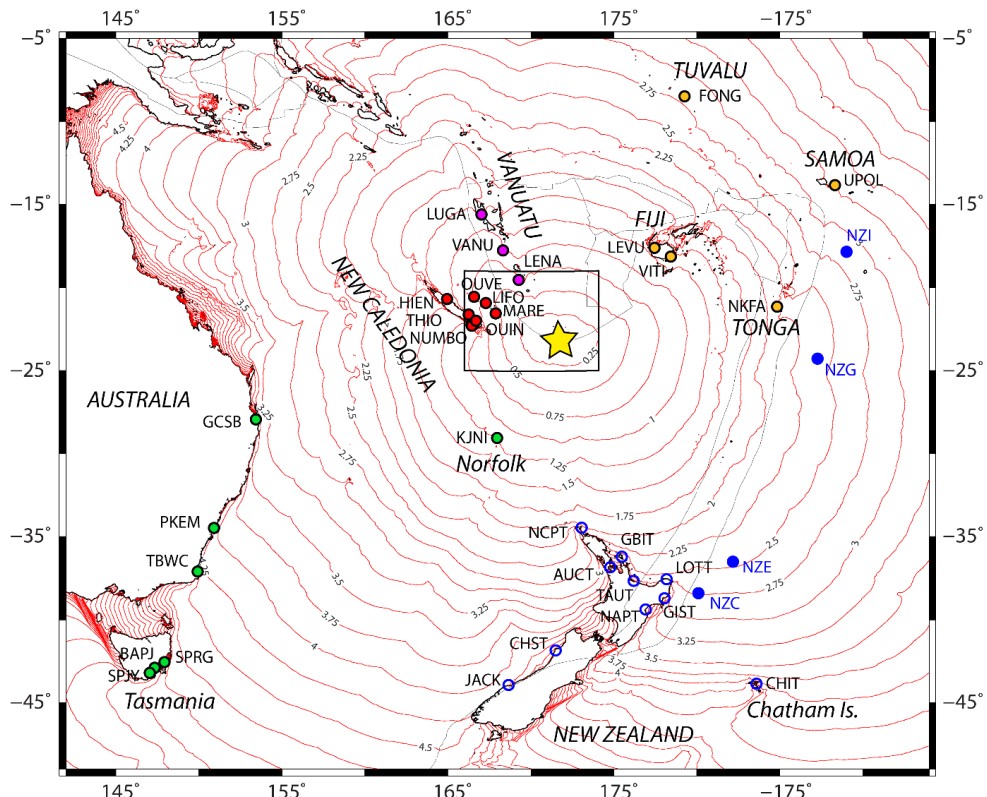

**Figure 5: Location of the coastal gauges having recorded the 10 February 2021 tsunami and computed tsunami travel times (TTT) at a regional scale (in hours). Coloured circles show the location of the stations (Blue: New Zealand – blue contour: coastal stations; full blue: DART stations; Green: Australia; Red: New Caledonia; Purple: Vanuatu; Orange: other countries) which recorded the tsunami; red lines represent the TTT isolines with a time step of 15 min; the yellow star locates the earthquake's epicentre; light grey lines represent the tectonic plate boundaries (GMT software dataset). The black rectangle locates the extent of figure 1.**





**Table 2.** Arrival times and amplitudes of the 10 February 2021 tsunami on DART stations and coastal gauges. They are classified from the first station (top row) recording the tsunami to the last one (bottom row).

| Station | Tsunami arrival time at station (UTC) | Tsunami travel time (hh:mm) | First wave amplitude (cm) | Maximum amplitude (cm) | Maximum amplitude time (hh:mm) | Delay between maximum and tsunami arrival time (hh:mm) |
|---|---|---|---|---|---|---|
| LIFO | 14:06 | 00:47 | 8 | 37.7 | 15:30 | 01:24 |
| MARE | 14:06 | 00:47 | 6.5 | 17.7 | 16:53 | 02:47 |
| LENA | 14:15 | 00:56 | 4.6 | 133.5 | 14:43 | 00:28 |
| OUIN | 14:26 | 01:07 | 17.6 | 27.9 | 15:05 | 00:39 |
| THIO | 14:34 | 01:15 | 7.1 | 9.8 | 18:02 | 03:28 |
| VANU | 14:38 | 01:19 | 0.2 | 4.9 | 15:22 | 00:44 |
| KJNI | 14:44 | 01:25 | 11.8 | 42.8 | 16:11 | 01:27 |
| HIEN | 14:47 | 01:28 | 2.5 | 9.6 | 16:52 | 02:05 |
| VITI | 14:49 | 01:30 | 4.6 | 4.7 | 15:35 | 00:46 |
| NUMBO | 14:55 | 01:36 | 0.8 | 2.4 | 16:38 | 01:43 |
| LEVU | 15:17 | 01:58 | 3.1 | 4.7 | 17:14 | 01:57 |
| NCPT | 15:26 | 02:07 | 2.5 | 28.8 | 16:51 | 01:25 |
| LUGA | 15:28 | 02:09 | 4.3 | 8.8 | 16:10 | 00:42 |
| NKFA | 15:29 | 02:10 | 3.3 | 3.6 | 18:49 | 03:20 |
| OUVE | 15:35 | 02:16 | 4.7 | 12.8 | 16:09 | 00:34 |
| NZG | 15:38 | 02:19 | 0.7 | 0.7 | 15:46 | 00:08 |
| NZE | 15:40 | 02:21 | 0.8 | 0.9 | 16:33 | 00:53 |
| LOTT | 15:58 | 02:39 | 6.2 | 24 | 17:09 | 01:11 |
| NZC | 16:00 | 02:41 | 1 | 1.4 | 16:22 | 00:22 |
| GBIT | 16:01 | 02:42 | 8.6 | 63.1 | 16:41 | 00:40 |
| NZI | 16:22 | 03:03 | 0.6 | 0.6 | 16:32 | 00:10 |
| TAUT | 16:24 | 03:05 | 0.7 | 4.2 | 21:10 | 04:46 |
| FONG | 16:25 | 03:06 | 2.4 | 3.8 | 17:16 | 00:51 |
| UPOL | 16:51 | 03:32 | 1.2 | 4.3 | 18:58 | 02:07 |
| GCSB | 17:15 | 03:56 | 15.6 | 30.2 | 17:30 | 00:15 |
| AUCT | 17:16 | 03:57 | 2.2 | 2.6 | 18:43 | 01:27 |
| TBWC | 17:35 | 04:16 | 3 | 9.5 | 19:05 | 01:30 |
| PKEM | 17:41 | 04:22 | 2.6 | 19.5 | 18:10 | 00:29 |
| CHIT | 18:04 | 04:45 | 2.2 | 7.7 | 21:39 | 03:35 |
| GIST | 18:05 | 04:46 | 0.7 | 6.6 | 20:39 | 02:34 |
| JACK | 18:26 | 05:07 | 1.4 | 36.2 | 21:45 | 03:19 |
| NAPT | 18:40 | 05:21 | 2.7 | 11.4 | 02:04 | 07:24 |
| SPRG | 19:02 | 05:43 | 1.4 | 7.3 | 20:02 | 01:00 |
| SPJY | 19:31 | 06:12 | 3.3 | 6.7 | 23:31 | 04:00 |
| BAPJ | 20:35 | 07:16 | 2 | 3 | 21:00 | 00:25 |
| CHST | | | | unidentifiable | | |
| SUMT | | | | unidentifiable | | |
| MNKT | | | | unidentifiable | | |
| PUYT | | | | unidentifiable | | |

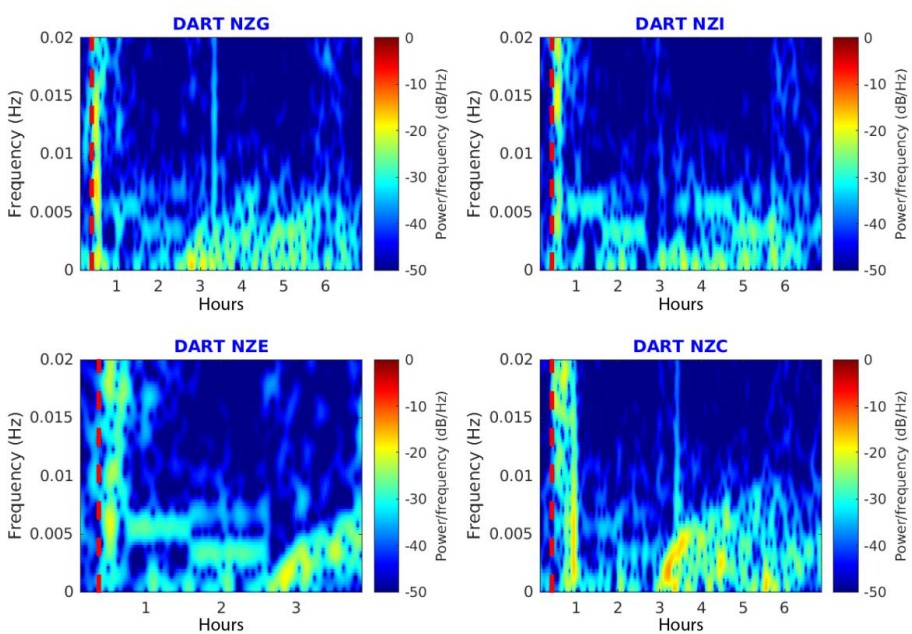

**Figure 6: Wavelet spectrograms for the 10 February 2021 Loyalty Island tsunami recorded on New Zealand DART stations. The red dashed lines symbolize the earthquake time.**

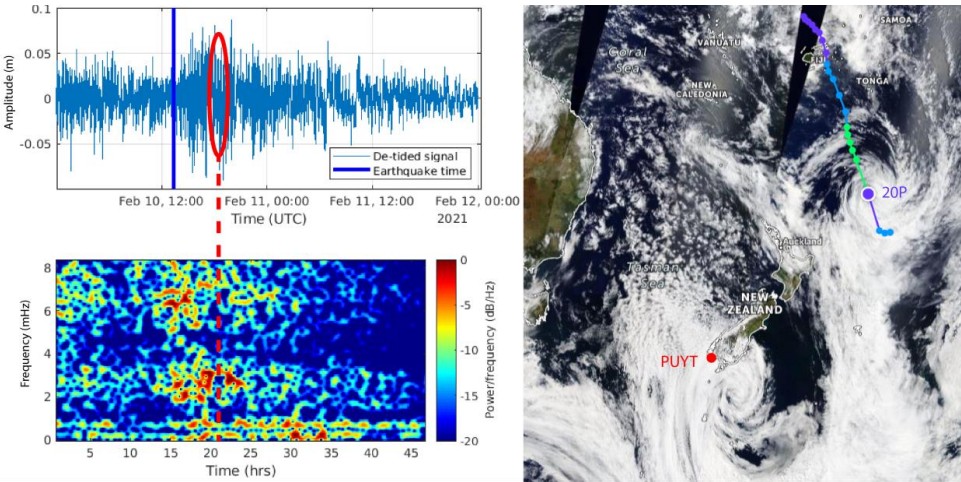

**Figure 7: Two storms on 10 February 2021 in the southwest Pacific Ocean. The south one is recorded by the Puysegur gauge (PUYT) at the predicted arrival time of the tsunami (red ellipse and dashed line) (Satellite image credits: Zoom Earth, NASA/NOAA/GSFC/EOSDIS, Suomi-NPP VIIRS).**



## 4. Tsunami numerical simulation

### 4.1 Methodology

The numerical simulations of tsunami generation and propagation for the four scenarios have been done using COMCOT (Cornell Multi-grid Coupled Tsunami model), a model progressively created during the mid-90s at Cornell University and then continuously developed at GNS Science, New Zealand, carefully tested and widely applied to numerous tsunami studies (e.g. Liu et al., 1995; Wang & Power, 2011; Wang et al., 2020). It computes tsunami generation, propagation and coastal interaction by solving both linear and non-linear shallow water equations using a modified explicit leap-frog finite difference scheme and considering the weak dispersion effect (Wang, 2008). The initial sea surface deformation is calculated using the Okada (1985)'s formulae with the fault plane geometry and either a uniform or non-uniform slip distribution. Water surface elevation and horizontal velocities are calculated respectively at the cell centre and at the edge centres of each grid cell of the computational domain. Absorbing boundary schemes are used at the boundaries of the computational domain to dampen the incoming waves, avoiding reflection from the grid boundaries.

For the purpose of this study, a set of nested numerical grids at different resolution levels has been prepared, covering the whole southwest Pacific region from 140 to 200°E and 0 to 50°S (first level grid 01) and specific areas (second level and its sub-level grids) focusing on each coastal gauge and DART station having recorded the 10 February 2021 tsunami and used in this study. Digital Elevation Models (DEM) used for these grids have been built from different datasets within the framework of previous projects. Norfolk Island high-resolution DEM has been specifically built for this study. The first level (grid 01) is at the lowest resolution (2 arc-min) covering the whole southwest Pacific region; its data comes from the ETOPO 1 global dataset (Amante and Eakins, 2009) with some refinements around New Zealand. The second level of grids, showing higher resolution of 30 to 24 arc-sec (~930 and 740 m respectively, cover several sub-regions focusing on New Zealand (grid 02), New-Caledonia/south Vanuatu (03), Norfolk Island (04), Australia east coast (Gold Coast – 05 and New South Wales - 06), Tasmania (07), Fiji (08), Raoul Island (09), Tonga (10), Samoa (11) and Tuvalu (12). Then, depending on the availability of higher resolution data, there is either one or two additional sub-level grids with



increasing resolution toward the area where a coastal gauge is located. The extent of most of the grids is presented on **Figure 8**. The resolution of each sub-level grid is calculated by COMCOT based on an input grid size ratio to the resolution of the previous level grid. The highest resolution used in this study is ~10 m in places where the bathymetry and the coastal shape is very complicated like Lenakel (Tanna Island, Vanuatu), as even minor inaccuracies in how these areas are represented could lead to very inaccurate results. For places like Tonga, Fiji, Tuvalu and Samoa where high-resolution dataset was not available for this study, virtual gauges have been positioned as closely as possible to the corresponding real gauge locations on the 30" resolution grids used for these places.

Tsunami waves propagation is subjected to linear, non-linear, and dispersion phenomena. As shown by Watada et al. (2014), the compressibility of the seawater, the elasticity of the solid Earth, and the gravitational potential variation associated with the mass motion during the tsunami propagation also play important roles on the tsunami travel times. These authors developed a method to correct the phase of the simulated waveforms, which has been applied to the synthetic time series obtained in the present study before comparing them to the recorded signals. The phase correction generally causes a slowdown of the tsunami and, by the way, reduces its amplitude.

*Note about the tides*

The southwest Pacific region tide dynamic is complicated, showing tide currents exceeding 5 cm/s in some places (Poulain and Centurioni, 2015) and New Zealand being at one of the amphidromic points, while showing large coastal tide amplitudes (Bye and Heath, 1975). It results in the tide pattern being drastically different from one side to the other of the Cook Strait (waterway separating New Zealand two main islands). Also, as some of the coastal gauges used in this study are located within a coastal lagoon (e.g. New Caledonia, Tonga, Fiji), it is worth noting that such semi-enclosed water body are also subject to specific tide behaviours, including amplification, delays, asymmetry of the tide fluctuations, and additional response to tidal oscillations (e.g. Albrecht and Vennell, 2007; Lowe et al., 2015; Green et al., 2018). These reasons lead to very different tide patterns and amplitude recorded on the gauges considered in this study as shown on **Figure 4**. To simplify the problem, it has been decided to simulate the tsunami propagation at mean sea-level (MSL) for each region without considering the
tide variations, although it has been shown that the tide-tsunami interactions can result also into important modification of the tsunami characteristics (amplitude and velocity mainly) in coastal zone (e.g. Kowalik et al., 2006; Kowalik and Proshutinsky, 2010; Zhang et al., 2011; Tolkova et al., 2015).

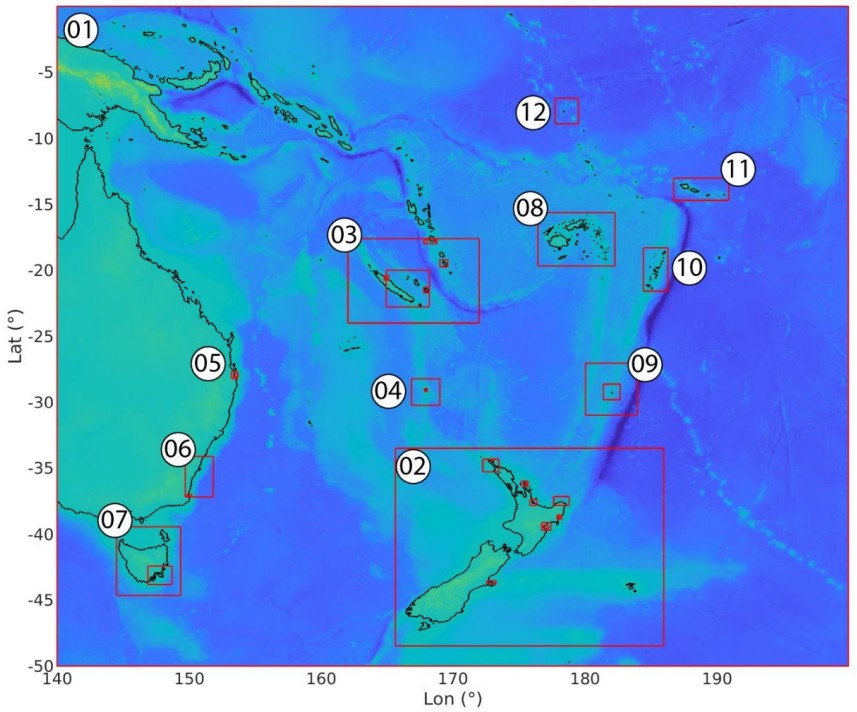

Figure 8: Extent of the grids used for modelling within the framework of the study. Grid 01 (1st level) covers the southwest Pacific region, from 140°E to 200°E and from 50°S to 0°, with spatial resolution of 2 arc-min. Numbers are associated to the grids of the second level with spatial resolution of 30 or 24 arc-sec. Higher resolution grids corresponding to additional levels are only indicated with red rectangles.

## 4.2 Results

The simulation results obtained with a uniform and two non-uniform slip models generally show good agreement with the recorded data either by the coastal gauges or the DART stations in the southwest Pacific region. A deep look at the results is necessary to highlight the differences and similarities between the three models. The results obtained with a maximum plausible $M_w$ 8.2 scenario are presented afterward.

### 4.2.1 Coastal gauge records



As shown on **Figure 5**, the 10 February 2021 tsunami has been recorded by at least 31 coastal gauges in the southwest Pacific Ocean. For the purpose of this study, and according to the quality of available bathymetric data, synthetic tsunami time series have been calculated at 24 of these 31 coastal gauges at the same locations or very close and compared to the real sea level data (**Figure 9**). The seven remaining gauges have not been considered because of the lack of quality bathymetric data at these locations. Generally, the simulated results are in good agreement with the real signals, in terms of travel time, amplitude, and polarity. Also, the wave patterns are very close from one scenario to another one in terms of first wave arrival time, general amplitude and polarity.

When looking into detail, it appears that the travel times difference between simulated and real records show a complicated pattern for each scenario, the simulations matching with the real tsunami arrival at gauges or being either too early or too late of 1-2 min with a maximum difference of 5 min. At LIFO, HIEN, NCPT, LUGA, OUVE, LOTT and GSB, the three scenarios first wave arrival matches with the real records. At VANU, VITI, FONG, TBWC and PKEM the three scenarios first wave arrival is too early. At LEVU, NKFA, UPOL, AUCT, JACK and SPRG it is too late. In the other locations, it is a mix between the three scenarios: at LENA and OUIN, scenario #2 matches the real records although it is too early for scenario #1 and too late for scenario #3; at THIO and KJNI, scenario #2 and scenario #3 match the real records although scenario #1 is too early.

Concerning the tsunami waves' polarity, the overall observation is that it is generally showing a good fitting of the first wave(s) considering the potential delay of the first arrival time. However, even if the following wavetrain is well correlated with the records, it sometimes shows a phase shift, associated with higher frequencies after the first hour of tsunami arrival.

Concerning the wave amplitudes, scenario #1 overestimates with a factor of 0.5 to 2 the first waves amplitude in near-field (LENA, OUIN, THIO, HIEN, VITI, LEVU) and northern New Zealand (LOTT, GBIT), although it fits it in further locations (KJNI, NCPT, LUGA, NKFA, OUVE). Scenario #2 fits correctly in near-field locations (OUIN, THIO, VITI, LEVU, OUVE), overestimates in near-field (VANU, HIEN) and in northern New Zealand (LOTT, GBIT) and lightly underestimates the wave amplitudes in most of the far-field locations (KJNI, LUGA, NKFA, FONG, GCSB, AUCT, PKEM,





CHIT, JACK, SPRG). Scenario #3 fits also in near-field locations (VITI, LEVU, OUVE) and in one far-field location (GBIT), overestimates in near-field locations (VANU, HIEN) and underestimates the amplitudes in nearly all other locations.

The non-uniform slip models (scenario #2 and scenario #3) show generally quite similar waveforms, scenario #3 being most of the time smaller than scenario #2 in terms of amplitude.

It is noticeable that the models are not able at all to reproduce the real signal at one location: VANU (Port Vila, Vanuatu) although numerous tests have been done to try to fit it correctly: changing the location of the virtual gauge, smoothing the bathymetric data or increasing its resolution. The other differences are related to the de-tiding method of the real signals using a polynomial fitting that is not always able to remove the whole components of the tide or to meteorological conditions like storm surges producing low frequency waves (e.g. SPRG and CHIT).

These comparisons need to be considered cautiously with regards to the overall small amplitude of the tsunami. But globally, scenario #2 presents a good compromise between the two other scenarios, being able to satisfy both near and far-field expectations. Thus, scenario #2 has been retained for further analysis presented hereafter.






**Figure 9: Simulation results obtained with 3 different seismic source model compared to 24 coastal gauge records: uniform slip model (red); non-uniform slip model from waveform inversion (yellow); USGS finite fault model (purple); real records (blue).**

### 4.2.2 DART records

Simulated sea level fluctuations due to tsunami waves at DART C, E, G and I location for each slip model are compared to real DART records on **Figure 10**. The reader must consider that the available 15 s sampling rate record transmitted in real-time by the BPR to the monitoring centre stops at 17:00 for DART E and stops at 18:30 for DART C, G and I.

In terms of arrival time, the three scenarios show good visual agreement with the records for the four stations. In terms of periodicity on each station, scenario #1 produces a leading wave period longer of 3-4 minutes than the records, leading to a phase shift of the wave train.

On DART C and E, scenario #2 provides the best match with recorded data in terms of arrival time and first wave amplitude and periodicity. A time shift of ~2 min occurs in the first through (after the leading wave arrival) and is reflected on the following waves, which is not the case with the scenario #3, fitting better the oscillations coming after the first wave.

On DART G, both non-uniform slip models provide a good match with the leading wave and then with the following with a small time shift of ~2 min.

On DART I, the three models seem to match correctly the tsunami waves, even if the interpretation of the results on such small amplitude signal of less than 5 mm must be done carefully.

To summarize, in terms of amplitude, the uniform slip model and the two non-uniform slip models are respectively slightly above or under the leading wave records within the range ± 2mm but generally show a good visual correlation between simulation results and records. Scenario #2 provides the best match for the leading wave without any surprise. The next few waves are better correlated with both non-uniform slip models in terms of amplitude and periodicity, the USGS model (scenario #3) showing a better fit with the oscillation and the other one (scenario #2) with the amplitude.

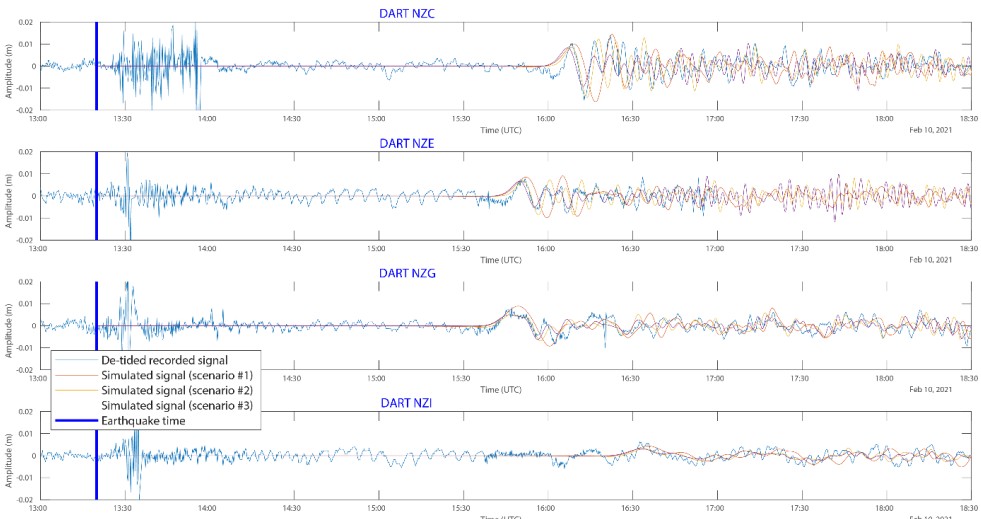

**Figure 10 : Sea level fluctuations associated to the 10 February 2021 earthquake and tsunami recorded by the New Zealand DART NZC, NZE, NZG and NZI: blue, red, yellow, purple lines represent respectively the de-tided real recorded data, the simulated signal for a $M_w$ 7.7 uniform slip model, the simulated signal for a $M_w$ 7.7 non-uniform slip model obtained from inversion of tsunami waveforms (Gusman et al., _in revision_) and the simulated signal obtained with the USGS $M_w$ 7.7 non-uniform slip model. The blue vertical line symbolizes the earthquake time.**

### 4.2.3 Maximum amplitudes

The maximum amplitudes maps presented on **Figure 11** and discussed hereafter are those obtained with the scenario #2.

At a regional scale, the maximum wave amplitude maps obtained after 12 hours of tsunami propagation over the southwest Pacific region show maximum amplitude not exceeding 1.5 m in the whole studied region, a main energy path oriented N-S (toward the north and west coasts of New Zealand and toward Tuvalu in the north) and strong bathymetric effect on the propagation (**Figure 11**). In fact, the presence of major bathymetric features of the mostly submerged Zealandia Continent (Mortimer et al., 2017) like the Lord Howe Rise and the Norfolk and West Norfolk Ridges (WN Ridge on **Figure 11**) between the source area and New Zealand/Australia and the numerous banks located in the north-west of Fiji, associated to the Vityaz trench, act as natural barriers and focus the tsunami south-westward and north-westward in specific locations outside of the earthquake region.



The role played by those submarine features in focusing the wave energy is clearly visible: Cape North in New Zealand and the south of New Caledonia, especially the Isle of Pines, respectively prolonging toward the south and the north the Norfolk Ridge which acts as a waveguide, are particularly exposed to tsunami waves. The Loyalty Islands Ridge and the Vanuatu subduction arc are acting as waveguides as well, focusing the tsunami waves towards the Loyalty Islands (Maré, Tiga, Lifou and Ouvéa) and the Vanuatu Islands (Aneityum, Tanna, Erromango, Efate mainly). This has already been highlighted by Roger et al. (2021) for the 5 December 2018 tsunami propagation. They are also two tsunami pathways clearly focusing the tsunami waves on Tasmania and along the Gold Coast (Australia). More locally, the tsunami shows relatively high amplitudes within lagoons and atolls like in Tuvalu, Tonga and Fiji or trapped around islands like around Norfolk or the Samoa Archipelago. It is to note that the tsunami is also amplified around the Chatham Islands, east of New Zealand. This could also be linked to the trapping of waves on the islands shelf. Finally, some places like Lenakel's Bay on Tanna Island, Vanuatu, or Jackson Bay on the southwestern coast of New Zealand are acting as "tsunami magnets", being able to catch tsunamis from a wide range of azimuths, and to show higher amplitudes of waves than nearby locations.

### 4.2.4 Plausible $M_w$ 8.2 scenario

The maximum wave amplitudes simulation of the tsunami triggered by a plausible Mw 8.2 earthquake rupture scenario proposed in this study are shown on **Figure 12**.

Without surprise, at a regional scale, the maximum wave amplitude maps obtained after 12 hours of tsunami propagation over the southwest Pacific region show maximum amplitude exceeding 0.5 m in many coastal zones of the studied region. The chosen strike of the fault rupture (287°N) is directly impacting on the orientation of the main energy path, NE-SW in that case (axis 17°-197°N), which needs to be considered cautiously: a slightly different strike would lead to a different orientation of the main energy path. Anyway, these simulation results underline much more the strong bathymetric role on the propagation. In the south of the trench, the main energy path is drastically deviated by the extension of the Loyalty Ridge south of the VSZ bending zone, leading to a propagation more perpendicular to the Norfolk Ridge, which seems to act as a barrier, with only one ray going through,



directly toward Lord Howe Island. Part of the energy is still propagating toward New Zealand, using the ridges like the Three Kings Ridge toward North Cape. To the north, the tsunami propagates within the North Fiji Basin, (between Vanuatu and Fiji) and is able to go through the Vityaz Trench region, reaching Tuvalu islands. Just a little part of energy is propagating toward the east and seems to disappear when crossing the Kermadec-Tonga Trench. In details, the tsunami seems to be caught within the different lagoons or trapped by shelf surrounding oceanic islands: Norfolk Island's shelf, for which a high-resolution DEM has been specifically built using nautical charts, is the best example of waves being caught around an island in this study, leading to consequent amplitudes of 1.5 m and more. High amplitudes are also shown in Vanuatu, especially on the southern coast of Aneityum Island, its southernmost island, but also in Tanna or Erromango, at the same locations already highlighted with the $M_w$ 7.7 scenario herein, but also for the 5 December 2018 tsunami study (Roger et al., 2021). In the nearby islands of New Caledonia, the amplitudes are less important as would have been expected, especially in the Loyalty Islands, but Ouvéa and Grande-Terre respective lagoons catch tsunami waves leading to amplitude records of around 0.5 m. Similarly, the tsunami waves are caught within the islands in Fiji, Tonga and in Tuvalu's Te Namo atoll. It is interesting to see that the tsunami can particularly affect the west coast of New Zealand much more than its northern shore: locations as Jackson Bay (southwest coast of the South Island), already identified as reacting very easily to tsunami coming from a wide range of azimuths, still shows amplitudes of more than 1 m.

**Figure 11: Maximum wave amplitude maps obtained after 12 hours of simulated tsunami propagation for the 10 February 2021 with a non-uniform slip model from waveform inversion (Gusman et al., in revision). The white circles locate the coastal gauges and DART stations having recorded the tsunami and used in this study. IdP: Isle of Pines; NI: Norfolk Island; NW Ridge: West Norfolk Ridge; VSZ: Vanuatu Subduction Zone (red dashed line).**



**Figure 12 Maximum wave amplitude maps obtained after 12 hours of simulated tsunami propagation for a plausible Mw 8.2 rupture scenario with uniform slip proposed in this study. The white circles locate the coastal gauges and DART stations having recorded the tsunami and used in this study. IdP: Isle of Pines; NI: Norfolk Island; NW Ridge: West Norfolk Ridge; VSZ: Vanuatu Subduction Zone (red dashed line). The white colour on the maps highlights zones where the tsunami amplitude reached over 1.5 m.**


## 5. Discussion

*5.1 Comparison of the slip models results*

The tsunami modelling results show that both uniform slip model built from CMT solution (scenario #1) and non-uniform slip models calculated from tsunami waves inversion (scenario #2) or seismic data (scenario #3) are able to reproduce the recorded signal of the small tsunami following the 10 February 2021 $M_w$ 7.7 thrust event generated at the southeasternmost part of the VSZ on most of the 24 coastal gauges and 4 DART stations of the southwest Pacific region considered in this study. This reproduction shows differences in some locations that can be attributed either to the resolution of the grids directly linked to the available bathymetric data quality, or to the dispersion phenomenon affecting the tsunami waves during propagation over long distances, or to the quality of the real coastal gauge data (including possible time and vertical offsets) or finally to the initial assumption on the source geometry used in tsunami inversion process.

This implies two things:

- a simple fault plane with uniform slip model (scenario #1) provides a good approximation of the amplitudes of a small tsunami on a set of DEMs focussed over the southwest Pacific region. This supports the results obtained by Roger et al. (2021) for the 5 December 2018 Loyalty Islands tsunami;

- we can use the first waves recorded at DART and coastal stations to produce a good estimation of the initial deformation (scenario #2) and use this initial (non-uniform) deformation to calculate the propagation on the whole region and confirm the related threat (for more information on the methodology, see Gusman et al., *in revision*). Depending on the relative location of the event epicentre to the stations' location, this could be done within a relatively short time using only the first 20-25 min of recorded tsunami waveforms. Considering that the New Zealand DART network is now fully operational with stations located close to the Hikurangi-Kermadec-Tonga and southern VSZ (three additional DART stations J, K and L have been positioned closer to the VSZ in July 2021), with detection capability of a tsunami within 30 minutes after an earthquake occurred in those 2 regions (Fry et al., 2020), it would

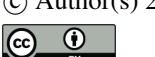



be possible to invert tsunami waveform with a good estimation of the initial surface
displacement within 50-55 minutes. This delay is unfortunately still too long to accurately
confirm the threat for neighbouring regions, e.g. for New Caledonia and especially the Loyalty
Islands and south Vanuatu if it occurs on the VSZ, but nevertheless in those specific cases it
can help for further exposed regions like New Zealand, the east coast of Australia, or
neighbouring Pacific Islands like Tonga, Fiji, Samoa, Tuvalu, Cook Islands and French
Polynesia. If it occurs in the southern VSZ like the 10 February event, it provides much more
time for New Zealand to confirm the threat by running inversion calculation. This inversion
methodology is interesting in the sense that it does not require a specific knowledge of the
geology of the source area.
*5.2 Role of submarine features*
This study particularly highlights the role of the mostly submerged Zealandia continent on the tsunami
propagation through the focusing and amplification of waves over particular submarine features. That
is probably why Lenakel Harbour (Tanna, Vanuatu) or Jackson Bay (New Zealand) have recorded
relatively important tsunami waves in comparison to neighbouring gauges. Concerning Vanuatu, this is
consistent with deaggregated hazard maps in probabilistic tsunami hazard assessments such as Thomas
and Burbidge (2009) who show that countries such as Vanuatu are exposed to tsunami hazard from the
entire VSZ (as well as the northern Kermadec-Tonga Subduction zone to a lesser extent) even if they
are not directly exposed.
It also highlights the trapping of waves around islands, especially around Norfolk Island, phenomenon
due to wave refraction and bottom-depth dependence on the island slope and shelf leading to the
development of oscillations of standing waves (e.g. Tinti and Vannini, 1995; Roeber et al., 2010; Zheng
et al., 2017). Resonance between islands probably needs to be considered to explain the wave
amplitudes observed in some archipelagos (Tonga, Fiji and Samoa) as explained by Munger and
Cheung (2008) for the 2006 Kuril Islands tsunami in Hawaii.





Finally, it reveals that some specific locations which seem to be protected from a tsunami generated at
the southernmost part of the VSZ like the Chatham Islands or Tuvalu can still be impacted.
*5.3 Contribution to regional hazard assessment*
The 10 February 2021 event brings new light on the ability of the southernmost part of the VSZ to
produce a regional event, being able to reach far-field locations as Tasmania in the south and Tuvalu in
the north, showing particular behaviours associated to submarine features and coastal shapes.
It is to note that this tsunami has not shown amplitudes like those of the 5 December 2018 tsunami
(from a $M_w$ 7.5 earthquake) on New Caledonia and Vanuatu coastal gauges because of its location
(further east), a different triggering mechanism (reverse faulting versus normal faulting) and the
direction of the main energy path (N-S instead SW-NE).
As the use of the model was validated with the $M_w$ 7.7 scenarios, it was the opportunity to look at what
would happen in the region if a tsunami was generated by a plausible magnitude $M_w$ 8.2 earthquake at
the southernmost part of the VSZ, which, as seen previously, has accumulated enough strain during at
least the last 100 years to be able to produce such event. According to the simulation results, the role of
waveguide and focusing of tsunami waves by submarine features of the former Zealandia continent
(limits from Mortimer and Scott, 2020) is enhanced, and a scenario of this type would have a greater
impact on the whole region in addition to all neighbouring islands of New Caledonia, Vanuatu and Fiji,
affecting the New Zealand north and west coasts and the east coast of Australia from the Gold Coast to
Tasmania as well. It would be of major interest to test many potential scenarios in the southernmost
part of the VSZ to see if they all behave the same way over those submarine features or not. The same
way, a set of scenarios would help to focus on very specific areas in the region that are prone to higher
tsunami amplitudes like Jackson Bay, Lenakel Harbor, Norfolk Island, etc., conducting high resolution
studies with a specific look at the resonance periods, and the wave trapping.
**6. Conclusion**
The 10 February 2021 tsunami triggered by a magnitude $M_w$ 7.7 earthquake at the southernmost part of
the VSZ has been recorded by at least 35 coastal gauges and DART sensors in the southwest Pacific



region. This small event is an additional opportunity to test the accuracy of the numerical model
COMCOT used for tsunami hazard assessment for New Zealand and find ways to improve the actual
warning system. The results of numerical simulations of tsunami propagation on a set of nested grids
of both uniform and non-uniform slip models presented in this study are able to reproduce the real
records with a relatively good correlation in terms of arrival times, wave amplitudes and polarity and
the identified differences could be linked to the lack of accurate bathymetric data in some places, to the
dispersion of the waves during the propagation, the potential bad quality of the real records and
eventually to the initial assumptions of the source location and geometry. As this event occurred in a
region where neither strong earthquake nor tsunami occurred during at least the last 100 years, the
validation of the $M_w$ 7.7 parameters for tsunami modelling will help to work on plausible scenarios
estimated for the southernmost part of the VSZ in agreement with geophysical data including the
subduction interface geometry which reproduce the curvature of the VSZ (SLAB 2.0: Hayes, 2018) and
look at their potential tsunami impact in the southwest Pacific region. It helps to highlight the significant
role played by the numerous submarine features in the region, focusing or stopping the tsunami waves,
the Norfolk Ridge being the most important acting like a waveguide toward the north and the south. It
also underlines the trapping of waves on Norfolk shelf and potentially around the Chatham Islands as
well. Finally, it highlights the difficulty to identify tsunami waves of small amplitude within a stormy
background.

**Data availability statement**
Most of the datasets used in the present study are available online: Global bathymetric dataset (ETOPO
1) is publicly available (https://www.ncei.noaa.gov/access/metadata/landing-
page/bin/iso?id=gov.noaa.ngdc.mgg.dem:316); high-resolution DEM covering New Caledonia and
Vanuatu has been prepared as part of the New Caledonia TSUCAL project and can be shared for
research purposes. Norfolk Island DEM has been specifically built for this project and can be shared
upon request. Other DEMs have been built in the framework of GNS Science research or commercial



projects and could be obtained under specific conditions (contact corresponding author for more
information). The earthquakes (https://earthquake.usgs.gov), centroid moment tensors
(https://www.globalcmt.org), coastal gauge records (https://www.linz.govt.nz/sea/tides/sea-level-
data/sea-level-data-downloads; http://www.ioc-sealevelmonitoring.org) and New Zealand DART data
(https://www.geonet.org.nz/tsunami/dart) are publicly available.

## Author contribution

JR has organized the study, performed the numerical simulations, analysed the results and written the
manuscript. BP has written the Tectonic context part and worked on the uniform slip scenarios
definition. AG has worked on the source from waveforms inversion and the analysis of the simulation
results and helped to write the manuscript. WP and DB have helped to improve the manuscript providing
constructive comments. XW has provided COMCOT code and assistance and worked on the
Methodology part. MD has prepared and provided high-resolution DEM for New Caledonia and
Vanuatu.

## Competing interests

The authors declare that they have no conflict of interest.

## Acknowledgments

This work has been funded by New Zealand's Strategic Science Investment Fund (SSIF). We would
like to acknowledge X and Y for reviewing this paper.

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
