# Peer review of "Potential tsunami hazard of the southern Vanuatu Subduction Zone"

_Natural Hazards and Earth System Sciences, 2022_

## Author Response (AR1)

**Dear Editor,**

**Please find hereafter our point-by-point response to the reviewers' comments.
Bold black writing corresponds to the online discussion comments and red writing
corresponds to what has been done/modified in the manuscript.**

**Sincerely,**

**Jean Roger et al.**

**\*\*\*\*\*\*\*\*\*\*\*\*\*\*\*\*\*\*\*\*\*\*\*\*\*\*\*\*\*\*\*\*\*\*\*\*\*\*\*\*\*\*\*\*\*\*\*\*\*\*\*\*\*\*\*\*\*\*\*\*\*\*\*\*\*\*\*\*\*\*\*\*\*\***

**RC1**: 'Comment on nhess-2022-157', Anonymous Referee #1, 28 Jun 2022 reply

 https://doi.org/10.5194/nhess-2022-157

**In the following we are answering (in bold writing) the comments/suggestions of the
reviewer RC1 (in italic writing):**

*The results of the proposed paper by Jean Roger & al; on the case study of the Matthew
Island tsunami of 10 February 2021 are relevant to be considered to assess the potential
tsunami hazard of the southern Vanuatu subduction zone.*

*The objectives of this paper are clear and reached. Nevertheless to assess the level of the
potential tsunami hazard, as indicated in the title of the paper, some modifications should be
considered by the authors.*

*The two main parameters considered in the tsunami threat and hazard are the estimated
arrival time of tsunami waves after the earthquake and the level of threat, directly related to
the tsunami height value observed or computed along the coastline, in particular at sea level
station location.*

*In this paper, several specific material are describing the tsunami threat in particular : figure
4 : tide gauge and DART records, table 2 arrival time and amplitudes ; Figure 11 and 12
Maximum tsunami height.*

*Several modifications would be needed to help to improve those figures, modifications that
will be considered separately (next chapter minor revisions and modifications).*

*Considering the tsunami hazard , it is internationally well known that several thresholds are
considered in tsunami hazard assessment and warning system : 30 cm for the first level of
threat, 1m for the second , 3 m for the third…*

*In table 2, it should be noticed that the maximum tsunami amplitude recorded is higher than
28 cm at 8 locations, and higher than 1 m at one location (Lena).*

- **Table 2 will be improved highlighting these 8 specific locations according to the
  international levels of threat as indicated.**

> ➔ **Done. As this is not international standard but more or less referring to a national choice, we don't mention international levels of threat, but "most standard warning level thresholds" (in part. 5.3)**

*The authors should highlight that this M 7.7 quake in that region resulted with a tsunami threat that need people evacuation for at least Height different sites, and probably many more (without tide gage records), considering the maximum amplitude modeled at Figure 11.*

- **A paragraph will be added to part 4.2.1 (Coastal gauge records) in agreement with this comment and the international levels of threat.**
  ➔ **Added in the discussion section (5.3)**

*On Figure 11 at each tide gage the maximum tsunami height records should be indicated on the map, circle with the scale color considering the value of maximum amplitude.*

- **White circles will be replaced with colored-scaled circles depending on the maximum amplitude of the tsunami at those locations.**
  ➔ **To do**

**Minor revisions and modifications:**

*L37 missing : tsunami height records of Mw 7,7 highlight that 30 cm tsunami waves amplitude were recorded at height different tide gages , included one raising more than 1m. The tsunami threat of that event should be considered for evacuation of the shoreline (coastal marine threat) at those locations.*

- **This sentence will be added in the abstract.**
  ➔ **Done**

*L43   add : **tsunami hazard, sea level records***

- **This is a good idea, the keywords will be added in the updated version of the manuscript.**
  ➔ **Done**

*Figure 1 : add one major seismic and tsunami event - yellow star -    Mw 7.0 : 19-11-2017*

- **We agree with the reviewer that the 19/11/2017 earthquake was one of the recent major events in the South Vanuatu subduction zone. Anyway, as shown by Roger et al., 2021, there have been several other tsunamigenic earthquakes of magnitude Mw < 7.5 occurring in the region. The objective of figure 1 was to highlight the seismotectonic context, without showing all the tsunamigenic events, but just the main ones, and the 1926 rupture having occurred east of the 2021 one. If we add the 2017 ones (3 earthquakes have been tsunamigenic during this crisis), we have to add a few others, leading to a reading problem of the figure. This explanation will also be added in the figure caption.**
  ➔ **Caption improved**

*L146 ... and **7.0**..*

- **Typo mistake; it will be corrected in the updated version of the manuscript.**
  ➔ **fixed**

*L198 which is generally **more accurate** ...*

- **"Better than" will be replaced with the reviewer's suggestion.**
  ➔ **modified**

*After P 9 Line number is missing !*

- **It is because line numbering breaks when there is a section break (next page) in the text. It will be fixed in the revised version of the manuscript.**
  ➔ **fixed**

*P10 : which is **1.6** smaller than those calculated*

- **"1.6" will be added in the updated version of the manuscript.**
  ➔ **added**

*P14 : Figure 4 : the blue line of the signal should be blue dark*

- **It will be changed.**
  ➔ **done**

*Several stations records have disturbances, in particular LIFO, LENA, GBIT, probably not related to tsunami waves*

*What is the origin of these sea level disturbances ?*

- **These sea level disturbances are certainly linked to the interaction of the tsunami waves with the semi-enclosed water body in which the coastal gauge is located. LIFO and LENA are located within small harbors, and GBIT is located within a bay. The period of the incoming waves can be equal or close to the harbor/bay eigenperiod and these could result in strong oscillations which represent a resonance behavior. Note that this phenomenon has already been discussed for LENA in Roger et al. (2021): the December 2018 tsunami led to strong oscillations in Lenakel's harbor.**
  ➔ **Added to the manuscript in section 3.3**

P16 : Table 2 :

*- how tsunami wave amplitude is measured ? Tide filtering, 0- crest; .....*

- **The explanation is briefly written in 3.3: "**The tsunami arrival times and amplitudes at each coastal gauge and DART station are summarized in Table 2. They have been obtained through de-tiding and filtering of the data using the methodology presented in Roger (Subm.)." **As the referenced paper is still under editor's decision, a description of the process will be added to the updated version of the manuscript, including the**

**type of filtering (bandpass) helping to remove both the tide signal and high frequencies related to other phenomenons like storms or large vessels. We measured the amplitude of the wave between 0 and the crest.**
→ **The referenced paper is still in the review process, so we updated the manuscript with a description of the process.**

*- for the stations LIFO, LENA, GBIT, due to the sea level disturbances, how did you measure the tsunami amplitude ?*

- **For the purpose of this study, we measured the largest amplitude of the whole signal: it means that the maximum amplitude indicated in Table 2 could be related to the resonance of the tsunami waves within a harbor and not the tsunami itself. In terms of risk management, it is worth it to consider the resonance as it could result in mooring breaks, whirlpools, enhancement of the inundation, etc. In fact, it is well known that catastrophic wave behaviors are often linked to nonlinear dynamics of which some resonances belong.**
  → **We don't consider it is necessary to add more information about this in the manuscript as it seems obvious that the maximum amplitude is the most important one in terms of hazard assessment, whatever it comes from resonance or not.**

*In particular at LENA, the behavior of the record (higher than the tide) would provide doubt about the accuracy of the tsunami amplitude measurement of the record.*

- **We agree that the appearance of the record in Lenakel is uncommon. An explanation would be that LENA tide gauge is located at the far end of a small bay, next to a concrete pier on one side and the mangroves on the other side (JICA report, 2013 - https://openjicareport.jica.go.jp/pdf/12129177.pdf) in a very small water depth (< 5 m according to nautical charts and probably much less according to J. Roger's own observation in 2019). Arrival of tsunami waves can results in a massive amount of water added in the bay rising up the mean water level (called wave setup), on which the next waves occur. This could lead to additional nonlinear behaviors and explain the records as shown on figure 4. Unfortunately, neither testimonials nor additional measurements have been collected after the event to validate the record with certainty.**
  **Only specific analyses of Lenakel's Bay behavior to a range of incoming waves would help to understand the process occurring during tsunami events, but also during storms but is off topic in this study.**
  → **We have added one sentence to specify this particular record in section 3.3.**

*P21 three scenarios first wave at too early… other it is too late. Authors should specify how early or late it is ( a few minutes… more ?) Are those delays negligible or not. Please specify.*

- **The time delay between simulations and records will be detailed.**
  → **Done**

*P23 Figure 9 the quality of the line of the records should be improved*

- **The quality will be improved in the updated version of the manuscript.**
  → **Done**

*P28 Figure 11 : at each tide gage, change the white color of the circle with the color corresponding to the maximum of the tsunami measured at the specific tide gage (maximum amplitude color scale)*

- **As previously indicated, the white circles will be updated with colored-scale circles according to the maximum amplitude.**
  - ➔ **The color of the circles has been linked to the maximum amplitudes recorded at gauge listed in Table 2**

*Change the color scale of the maximum amplitude : the current color scale is green from 5 cm to 25 cm. This scale color is not helping to visualize where and how the tsunami height is growing from 5 to 25 cm : the largest surface of the sea.*

- **Many tests of colors have been tried, but this was the most relevant we found. Anyway, we will try to update it a better way to underline the amplitude differences.**
  - ➔ **Done**

*P29 Figure 12 same remark as figure 11 regarding the maximum amplitude scaling color.*

- **Same answer as previous point.**
  - ➔ **Done**

*P32 Contribution to regional tsunami hazard assessment*

*The comments and proposed modification made above on the threat level should be added in that chapter.*

- **This will be added.**
  - ➔ **Added in 5.3.**

*P33 The results of the tsunami hazard assessment of the 8.2 scenario should be added, and in particular that such earthquake would generate tsunami waves height at shoreline higher than 1 m in many places at New Caledonia, Vanuatu, Fiji, New-Zealand, ...*

- **That is true, it should be added in the conclusion.**
  - ➔ **Added.**

*P34 L 112 : and **wrote** the...*

- **This grammar comment will be considered during the review of the English-native co-authors.**
  - ➔ **Fixed and reviewed**

\*\*\*\*\*\*\*\*\*\*\*\*\*\*\*\*\*\*\*\*\*\*\*\*\*\*\*\*\*\*\*\*\*\*\*\*\*\*\*\*\*\*\*\*\*\*\*\*\*\*\*\*\*\*\*\*\*\*\*\*\*\*\*\*\*\*\*\*\*\*\*\*\*\*\*\*\*\*\*\*\*\*

**RC2**: 'Comment on nhess-2022-157', Anonymous Referee #2, 04 Jul 2022 reply

**In the following we are answering (in bold writing) the comments/suggestions of the reviewer RC2 (in italic writing):**

*Overall, a nice paper describing the potential of the region to produce moderate tsunamis, the typical result of such tsunamis, and a comparison of the recent tsunami with modeled results. I would recommend this paper be published with some minor revisions. My criticisms and suggestions follow:*

*Since source scenario #2 (inverted from DART and gauge waveforms) is the one determined to best fit the data and is singled out in the discussion section, some description of how it was obtained would be nice as the Gusman et al publication has not been published yet. Particularly since good inversions from coastal gauges have historically been difficult to produce due to the fact that nonlinear effects become more important in the shallow bays and coastlines where tide gauges are typically installed. It's been the subject of enough debate that simply referencing an unpublished manuscript is not quite sufficient here (though perhaps it would be if it weren't chosen as the featured source in this paper). Also, no figure showing the slip distribution is offered, nor a figure of the resulting dislocation. Nothing in the way of how much data were used or why coastal gauges can be used in this case, or whether tides were inverted with the data, or detided and then inverted. Not to say the inversion is not a good one, but that inversions with coastal gauges has not always been too successful and this source is the one picked out as best for this event. Please provide a little more info on how the inversion was produced and perhaps a figure of the slip distribution.*

- **Concerning the slip distribution model from waveforms inversion, the other manuscript is currently with the editor for a decision. Hopefully we could have the other manuscript accepted soon. To prevent a dual publication, we would like to refer to Gusman et al. for the slip distribution which is already available as a proof (https://www.essoar.org/doi/10.1002/essoar.10507385.1). We can provide a plot for the slip distribution as requested by the reviewer.**
  - ➔ **As the referred paper has been published, the reference has been updated and we have decided to only refer to it to prevent dual publication. The slip distribution figure is shown in this paper on Figure 4 p.7**

*Regarding the phase of the time series in the paper, page 19, 2nd paragraph, "These authors developed a method to correct the phase of the simulated waveforms..." do you mean "The authors" (yourselves) or the authors of Watada, et al? In either case, please elaborate briefly: were the phases adjusted manually or by some computational method devised by Watada? You state that the phase-change method reduces amplitude - do you find that overall modeling results underestimate due to this phase reduction? This seems important to clarify because you are, after all, judging the sources in the paper largely by the accuracy of the modeled time series.*

- **The method to correct the simulated waveforms has been developed by Watada et al. (2014). We would like to clarify that we have calculated the phase correction with a computer code but not manually. We could provide another plot showing the simulated waveforms with and without phase correction at a selected DART station if requested. More details will be provided in an updated version of the manuscript.**

➔ **Paragraph has been improved and new references added.**

*Lastly, the choice of the 3rd source in the overall study of regional hazard assessment addressed in this paper supposes, rationally, that if the 2021 event is Mw 7.7, that a larger one may occur in the future. The question becomes why did the authors choose Mw 8.2 as an appropriate maximum for the region? You cite a range of magnitudes from Ioualalen et al (2017), Gutenberg (1956), Richter (1958) and Engdahl and Villasenor (2002), but why choose 8.2 specifically? Did I miss an estimate of rupture length limit, or strain rate? Or perhaps is Mw 8.2 not implied as the maximum for this region of this subduction zone? Simply make it clear that this is an estimate of the maximum along this section of the fault and why.*

- **This magnitude Mw 8.2 scenario has been built using:**
  - **The estimation of maximum magnitude of Mw8.1-8.2 for the 1875 South Vanuatu earthquake by Ioualalen et al. (2017)**
  - **The fact that the earthquakes location and the calculated moment tensor solutions (USGS and GCMT) provide an available subduction zone length (~250-300 km) in the south of the VSZ which can accommodate a Mw8.2-8.3 according to the empirical relationships from Strasser et al. (2010)**
  - **The maximum value in the USGS earthquake catalogue for the Vanuatu subduction zone is Mw8.1 for the 21 Sep. 1920 earthquake.**
- **A scenario following the curve of the VSZ and going on toward the north would produce a larger magnitude but**
  - **We don't know if the rupture would be able to go through the area where the Loyalty Ridge is subducted**
  - **The aim of the study was to discuss the impact of one larger realistic case at a regional scale instead of testing all possibilities**
- ➔ **Paragraph has been improved to provide more information to the reader.**

The following comments I hope will make the paper a little more clear. My apologies if I criticize unnecessarily: I will try not to suggest changes that only affect tone and do not detract from the science.

*ln 48, name change from New Hebrides SZ to Vanuatu SZ: my question "who gets to name these things"? Call it a chocolate lollipop for all I care, but is VSZ the generally-accepted replacement for NWSZ? Why did it change? If you mention it at all ("...former New Hebrides Subduction Zone...") then perhaps noting why it changed would please the reader.*

- **We decided to change the name in a previous paper about the Tadine tsunami of 5 December 2018. "New Hebrides" is related to the colonial times of what became "Republic of Vanuatu" in 1980. This change respects both this political change, and the population of Vanuatu amongst which we have colleagues and friends.**
  - ➔ **We decide not to add more information about this as it is well-known that New Hebrides has been replaced by Vanuatu 40 years ago.**

*ln 81, convergence rate "in the northern part" are stated as 16-17 cm/y, but Figure 1 white arrow only shows 12 cm/y. If the larger value is farther north than the figure shows, then perhaps mention it?*

- **It will be mentioned in an updated version of the manuscript.**
  - ➔ **Added to the manuscript on line 81**

*ln 118, ah I see, you note that the 12 cm/y is the "southern part of the VSZ". Perhaps mention that the 16-17 cm/y values are outside the the figure 1 extents?*

- **See previous comment**
  - ➔ **Added on line 81**

*ln 106, is the word "crises" a seismic term?*

- **Yes, it is.**
  - ➔ **Nothing to modify here**

*pg 13, 2nd paragraph "...DART station relatively to the strike...": change to "relative to the strike"*

- **It will be changed in the updated version of the manuscript.**
  - ➔ **changed**

*Figure 4, some gauge arrival time blue lines are too thin to see (OUIN), and some don't show an obvious wave (LEVU), though sometimes this can be hard to determine and can be dwarfed by the tidal amplitude on the plot. Consider using a thicker blue line?*

- **The lines will be redrawn thicker.**
  - ➔ **Same comment as RC1; the figure has been improved.**

Figure 9: the lines are so thin that I can almost not tell the difference in color between yellow and red. Please make these thicker even if it masks some high-frequency oscillations. For some reason Figure 10 is much easier to read.

- **As suggested by the reviewer RC1, the lines will be redrawn thicker.**
  - ➔ **Figure improved with thicker lines**

*Figure 10 caption: don't use "respectively" for color-coding: it is confusing. Simply list each source and put the color in parenthesis after OR (since you have a legend) use the source number like so: "the simulated signal for a Mw 7.7 uniform slip model (source #1)", etc*

- **Good idea, the caption will be modified according to this comment.**
  - ➔ **Caption modified**

*Page 27, last sentence: good point about the west coast of New Zealand being susceptible to tsunami, but the word "still" implies that waves are high despite this event, not because of it. Consider "also shows amplitudes of more than 1 m."*

- **The sentence will be improved according to the comment.**
  - ➔ **modified**

*Page 31, lines 44-46: the authors state that "...Vanuatu [is] exposed to tsunami hazard ... even if they are not directly exposed". I think the meaning is that Vanuatu is exposed to high*

*tsunami hazard even if the main wave energy of a given tsunami does not directly focused at Vanuatu?*

- **Absolutely, we also think that the sentence is hazardous, we will rephrase it in a more understandable way.**
  ➔ **Sentence improved**